# Meta-analysis of QTLs associated with popping traits in maize (*Zea mays* L.)

**Sukhdeep Kaur[1], Sujay Rakshit [2]\*, Mukesh Choudhary[2], Abhijit Kumar Das[2], Ranjeet Ranjan Kumar[3]**

**1** Department of Plant Breeding & Genetics, Punjab Agricultural University, Ludhiana, India, **2** ICAR-Indian Institute of Maize Research, PAU Campus, Ludhiana, India, **3** Division of Biochemistry, Indian Agricultural Research Institute, Pusa, New Delhi, India

\* s.rakshit@icar.gov.in

**Data Availability Statement:** All relevant data are within the paper and its Supporting Information files.

**Funding:** The authors received no specific funding for this work. However, the work was carried out

## Abstract

The rising demand for popcorn necessitates improving the popping quality with higher yield of popcorn cultivars. Towards this direction several Quantitative Traits Loci (QTLs) for popping traits have been identified. However, identification of accurate and consistent QTLs across different genetic backgrounds and environments is necessary to effectively utilize the identified QTLs in marker-assisted breeding. In the current study, 99 QTLs related to popping traits reported in 8 different studies were assembled and projected on the reference map "Genetic 2005" using BioMercator v4.2 to identify metaQTLs with consistent QTLs. Total ten metaQTLs were identified on chromosome 1 (7 metaQTLs) and 6 (3 metaQTLs) with physical distance ranging between 0.43 and 12.75 Mb, respectively. Four identified metaQTLs, *viz.*, mQTL1_1, mQTL1_5, mQTL1_7 and mQTL6_2 harboured 5–8 QTL clusters with moderately high $R^2$ value. The clustered QTLs were from two or more experiments. Based on the expression pattern in endosperm and pericarp tissues, a total of 229 genes were selected. Nineteen of these genes are involved in carbohydrate metabolism. Of the 19 genes specifically involved in carbohydrate metabolism, 11 of them were in these regions, implying the importance of these clustered QTLs. MetaQTL1_1 at bin location 1.01 coincided with the reported QTLs related to various agronomic traits like stalk diameter, tassel length, leaf area and plant height. The identified metaQTLs can be further explored for fine mapping and candidate gene identification, which can be validated by loss or gain of function. Identified metaQTLs can be used for introgression of popping traits towards enhancing the popping ability.

## Introduction

Maize (*Zea mays L*) is one of the most versatile crops in the world with diverse utilization. Six major types of maize, *viz.*, dent corn, pod corn, flour corn, sweet corn, flint corn and popcorn prevail in the world. Specialty corns such as sweet corn, baby corn and popcorn add new avenues of economic returns to maize farmers. Popcorn expands and puffs up upon heating, and is one of the favorite pastime foodstuffs with high nutrition value [1]. Popcorn is a type of flint

under general financial assistance from the Indian Council of Agricultural Research, New Delhi (India) to support research activities key in carrying out this work.

**Competing interests:** The authors have declared that no competing interests exist.

corn, modified by the selection process to maximize popping expansion [2]. The current value of the Global Popcorn market is around 3794.1 million US$ which is expected to grow to 5550 million US$ by 2025 with a Compound Annual Growth Rate (CAGR) of 6.7% (www.marketwatch.com). Worldwide one of the largest manufacturers of microwave popcorn is ConAgra Foods which supplies in more than 30 countries. In India also, there is a huge industrial demand for popcorn which is evident from the fact that despite a heavy customs duty (56%), the two major popcorn industries, *viz.*, Agro Tech Food and BanacoOverseas annually import around 23,000 tonnes of popcorn (brandgiri.wordpress.com). The emerging demand for popcorn attracted the researchers and investigators to increase the popping ability for fulfilling the consumer satisfaction [3]. Popping trait is characterized as the function of physical structure of kernel and endosperm composition. Though popping is also reported in normal dent and flint corn, the flakes are too small as compared to standard popcorn. The size of flakes produced depends upon the ratio of hard to soft endosperm in the popcorn cultivar [3]. The popping also depends upon the optimally applied temperature and resultant pressure. Any deviations from optimal conditions of temperature and pressure result into poor popping. When the kernels are heated to about 180˚C, the water (grain moisture) expands leading the pressure to rise to 135 pounds per square inch (psi). The resultant high pressure leads the hard pericarp to burst open [3]. The starch granules get gelatinized and inflated by heat forming a 3D structure [4].

There are number of other factors which affect the popping in popcorn. First is the pericarp thickness, which is important to build up pressure inside kernel up to certain point in order to get good sized flake. Though dent × popcorn crosses carry thick pericarp but on explosion it gets shattered into big pieces leading to poor eating grades. Thus, the breeders aim at selection for popcorn genotypes with thin pericarp [4]. Second factor is flake phenotype which play vital role in consumer preference and processing. Two distinct types of flakes are produced in popcorn, *viz.*, butterfly and mushroom type. Former one breaks very easily during processing or storage, leading to low popping volume and dull appearance, whereas mushroom type is round and not prone to breakage, thus easily amenable to packaging and value addition. Butterfly type is the most common type popcorn preferred by the consumers due to more tenderness and freedom from hull. Mushroom type is highly preferred by vendors but less common due to non-availability of 100% mushroom type flakes producing lines commercially. Third factor is related to agronomic management practices. Popcorn genotypes are less vigorous and require agronomic operations to speed up emergence and initial vigour. The optimal moisture content at the time of harvesting should be 16–18% to avoid mechanical damage to pericarp as well as maintenance of popping ability [4]. Popping Expansion Volume (PEV) is one of the important traits that distinguishes popcorn from other corn types and is an important target trait for popcorn breeding. There are several other traits those influence popping quality as detailed in Table 1. Other than PEV, popping rate (PR) or percent unpopped kernel (UPK), flake volume (FV), flake size (FS) and average kernel size (AKS) are the most important traits influencing popping quality, while others like starch concentration (CT), protein concentration (CP) and oil concentration (CF) indirectly influence popping, and grain weight (GW) determine popcorn yield. Investigators used correlation and path coefficient analysis to measure direct and indirect effect of independent traits on dependent ones like popping expansion. Flake volume (FV) and percent unpopped kernel (UPK) are positively and negatively correlated with PEV, respectively [5], whereas AKS is negatively correlated with PEV, indicating the importance of small seed size for better popping expansion [1,5]. Significant high correlation among PR, PF and popping volume (PV) has also been reported [6]. The trait CP has been found to be positively correlated with PV and FS, whereas CT and CF were negatively correlated to flake size [6]. Popcorn germplasm has narrow genetic base as most of the popcorn

**Table 1. Popping related traits and their details commonly used in popocorn breeding with abbreviations used in the study.**

| Trait | Remarks |
|---|---|
| Popping expansion volume (PEV) | Popped corn volume/original corn volume |
| Popping volume (PV) | Popped volume/100 kernel unpopped volume |
| Popping rate (PR) or percent unpopped kernel (UPK) | Number of popped or unpopped kernels per total number of kernels taken |
| Flake volume (FV) | Absolute volume of 100 popped kernels |
| Flake size (FS) | Total popped volume/no. of popped kernels |
| Popping fold (PF) | Popped volume/100 kernel weight total |
| Starch concentration (CT) | Measured using MATRIX-1 NIR (near infrared reflectance) spectroscope |
| Protein concentration (CP) | Measured using MATRIX-1 NIR spectroscope |
| Oil concentration (CF) | Measured using MATRIX-1 NIR spectroscope |
| Average kernel size (AKS) | Average face area of each kernel |
| 100 grain weight (100GW) | Weight of the randomly taken 100 grains |
| Grain weight per plant (GWP) | Weight of grains per plant measured in grams |

lines are derived from the selected flint germplasm [7]. There is opportunity to diversify the narrow gene pool of popcorn using normal maize germplasm but linkage drag, change in allele expression in the new genetic background and lack of recovery of favorable allelic combination in segregating populations hinder progress in popcorn breeding. Efforts to improve the popcorn gene pool using dent maize although helped to improve agronomic traits, *viz.*, seed weight, yield, disease resistance, insect resistance and stalk strength, but at the cost of drastic reduction in PEV [3,8]. Hence, the availability of molecular markers and knowledge about the genetic architecture of the popping traits can address all the challenges in popping improvement programme [9].

Most of the popping traits are quantitative in nature and highly influenced by environment (E) and genotype (G) × environment (E) interaction effect. QTL mapping is one of the potential approaches to locate the genomic regions of quantitative traits like PEV and other popping traits. A number of major QTLs explaining over 10% phenotypic variance (PVE), associated with various popping traits have been reported in earlier studies [5,6,10–13]. The outcome of different QTL mapping experiments is affected by various factors including environment, cross type, size of mapping population and type of markers used the experiments. This leads to difficulty to conduct marker assisted selection (MAS) by directly using these reported QTLs. Meta-analysis paves the way to identify the true QTLs by sequentially combining QTLs reported from different studies, developing the consensus map and validating the consistent QTLs from one experiment to another. Akaike information criterion (AIC) based method is used to determine number of actual QTLs constituted by the QTLs discovered in different studies. Simulations are executed to study the quality of the model obtained using AIC. The consistent QTLs (at least two) identified by meta-analysis for a set of QTLs at a confidence interval (CI) of 95% using independent studies is referred to as meta-QTL [14]. The meta-QTLs meeting certain criteria, *viz.*, a steady and large effect on target trait(s), smallest CI and clustering of high number of initial QTLs, are to be selected for marker assisted selection [15]. These regions with precise positions are more accurate and subjected to candidate gene mining from available physical map. Functional analysis of the identified candidate genes can further narrow down to actual gene(s) which are directly or indirectly contributing to traits of interest.

In this study, the meta-analysis was carried out for QTLs related to popping traits to address the query whether these studies could be used to build a consensus map and to project meta-QTLs on it by means of iterative projection step using BioMercator v4.2 [16]. Another objective was to find the flanking markers of those meta-QTLs with small CI which can be possibly used in MAS.

## Materials and methods

### Literature review and QTL data compilation

Published works on QTL mapping for popping traits in maize were identified through database search using the keywords, *viz*., popcorn, QTL, metaQTL, popping expansion volume, popping volume, and surveyed in google search engine and PubMed. Ten reports were available regarding QTL analysis of the traits under study. The studies where QTLs were identified in two different mapping populations were considered separate experiments. Out of ten reports two could not be used because the used primers were coded (instead of marker names), in addition SNP markers were used to construct the maps which could not be plotted to the consensus SSR map because of uncommonness of markers (Fig 1). The eight reports used in this study had three different mapping populations ($F_{2:3}$, $BC_2F_2$ and RIL) derived from two crosses, *viz*., (Dan232 × N04) and (A-1-6 × V273). The data from different studies regarding parents, kind of mapping population, cross size and type of markers used were put together into one table (Table 2). For the analysis purpose, information from individual (published) studies was organized into two different files-one for QTL data (.qtl) and the other for the position of genetic markers (.map). In addition, experiments (.txt) file was also created that contained the overall information about the experiments used to metaQTL analysis. Reported QTLs along with $R^2$ values in respective studies used in metaQTL analysis are represented chromosome wise in Fig 2. Other parameters included in QTL data file are QTL name, group, position, LOD value and CI (reported in respective papers). In case where the positions of QTLs were not provided the averages of lower and upper CI of the QTLs were considered. QTL-map (.qtl), genetic map (.map), reference map (.txt), experiments (.txt) and trait_ontology (.txt) files were used to generate XML files using MetaQTL software (http://bioinformatics.org/mqtl). The generated XML files were further used as input files in the BioMercator v4.2 for meta-analysis process.

### Map projection and consensus map integration

The map and QTL files were uploaded in BioMercator v4.2. It connects together the input files and checks for the common marker sequences between each pair of input maps. Using Bio-Mercator v4.2 software, InfoMap step was performed to check the connection between the maps of different experiments. Later, the collected QTLs for popping traits were projected on reference map "Genetic 2005" with more than 2000 markers using BioMercatorv4.2 in order to get metaQTLs and refine CI of the QTLs. Chromosomal regions containing only one QTL were discarded from the analysis as by definition meta-analysis involves two or more consensus QTLs. Iterative map projection was carried out to project markers and QTLs from individual genetic maps on the reference map starting with the map having the highest resemblance to the reference map. This consensus step integrates all the maps in a single step using a weighted least square strategy.

### QTLs projection

To project QTLs on consensus map LOD score, $R^2$ value, flanking marker positions and CI were utilized. The locations of QTLs were detected using simple scaling rule between the initial

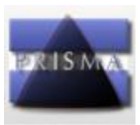

## PRISMA 2009 Flow Diagram

**Identification**

Records identified through
database searching
(n = 8 )

Additional records identified
through other sources
(n =  0)

Records after duplicates removed
(n = 8)

**Screening**

Records screened
(n = 8)

Records excluded
(n = 2)

**Eligibility**

Full-text articles assessed
for eligibility
(n = 6)

Full-text articles excluded,
with reasons
(n = 0)

**Included**

Studies included in
qualitative synthesis
(n = 8)

(The studies where QTLs were identified in two
different mapping populations were considered
separate reports)

Studies included in
quantitative synthesis
(meta-analysis)
(n = 8)

*From:* Moher D, Liberati A, Tetzlaff J, Altman DG, The PRISMA Group (2009). *P*referred *R*eporting *I*tems for *S*ystematic Reviews and *M*eta-*A*nalyses: The PRISMA Statement. PLoS Med 6(7): e1000097. doi:10.1371/journal.pmed1000097

**For more information, visit www.prisma-statement.org.**

**Fig 1. Flow diagram of studies assessed and included.**

**Table 2. Details of mapping studies used for popping traits in meta-QTL analysis.**

| Sl. No. | Map | Cross name | Mapping population | Marker Type | Cross size | QTLs present | Reference |
|---------|-----|-----------|--------------------|-------------|-----------|--------------|-----------|
| 1 | E1 | A-1-6 × V273 | $F_{2:3}$ | SSR | 194 | 13 | [5] |
| 2 | E2 | Dan232 × N04 | $F_{2:3}$ | SSR | 259 | 24 | [12,13] |
| 3 | E3 | Dan232 × N04 | $F_{2:3}$ | SSR | 259 | 27 | [10,17] |
| 4 | E4 | Dan232 × N04 | $BC_2F_2$ | SSR | 259 | 16 | [10,17] |
| 5 | E5 | Dan232 × N04 | RIL | SSR | 258 | 19 | [6] |

[#]$F_{2:3}$: $F_2$ derived $F_3$ families; $BC_2F_2$: Backcross derived $F_2$ population; RIL: Recombinant inbred lines.

QTL marker distance and the corresponding distance on the consensus map. Approximation of the new CI for projected QTLs was carried out with Gaussian distribution encompassing the most probable QTL position. All the QTL projections were conducted using BioMercator v4.2.

## QTL meta-analysis

Meta-analysis requires the presence of independent QTLs for a particular trait obtained from different mapping populations, different locations or different environments. On that basis of an integrated consensus map and initial QTL projections, meta-analysis step was performed on QTL clusters present on each chromosome using BioMercator v4.2. In this method, all the possible QTL combinations were tested and the one which maximizes the likelihood was selected. Two steps in meta-analysis were followed. In step 1, QTLs on each linkage group were clustered, assuming their normal distribution around the true location. Subsequently, the QTL model on each linkage group was selected using the Akaike Information Criterion (AIC). The model with the lowest AIC represents the number of meta-QTLs. In step 2, meta-QTLs were generated in accordance with the best model. Further, the position and CI (95%) of the meta-QTLs were calculated and the flanking markers for meta-QTLs were selected. The locus lookup browser was used to determine the physical position of the flanking markers (www.maizegdb.org). In cases where the markers physical positions were not mapped, the next closest outer marker was used. The physical length of identified meta-QTLs was characterized to retrieve candidate genes linked with popping from the maizeGDB database (http://maizegdb.org/). Further, 'qTeller' tool available on maizeGDB was used for identification of genes related to endosperm and pericarp tissues in all the identified meta-QTL regions (https://qteller.maizegdb.org/). The candidate genes sequences were aligned to the Kyoto Encyclopedia of Genes and Genomes (KEGG) pathway database using the online KEGG Automatic Annotation Server (KAAS) (http://www.genome.jp/kegg/kaas/).

## Results

### Nature of reported QTLs for popping traits

A total of 99 QTLs, from studies related to popping traits, distributed across ten chromosomes of maize were used in the present study. The highest numbers of 33 QTLs were present on chromosome 1, whereas least number of QTLs (2) were found on chromosome 9 (Fig 2). Out of total 99 QTLs, 67 (67.67%) illustrated $R^2$ value of less than 10%. It signifies that small proportion of phenotypic variance is explained by these QTLs and hence, considered as minor QTLs. However, rest of 32 QTLs (32.33%) with $R^2$ value of greater than 10% were considered as major QTLs. This leads to conclusion that popping traits in maize are controlled by large

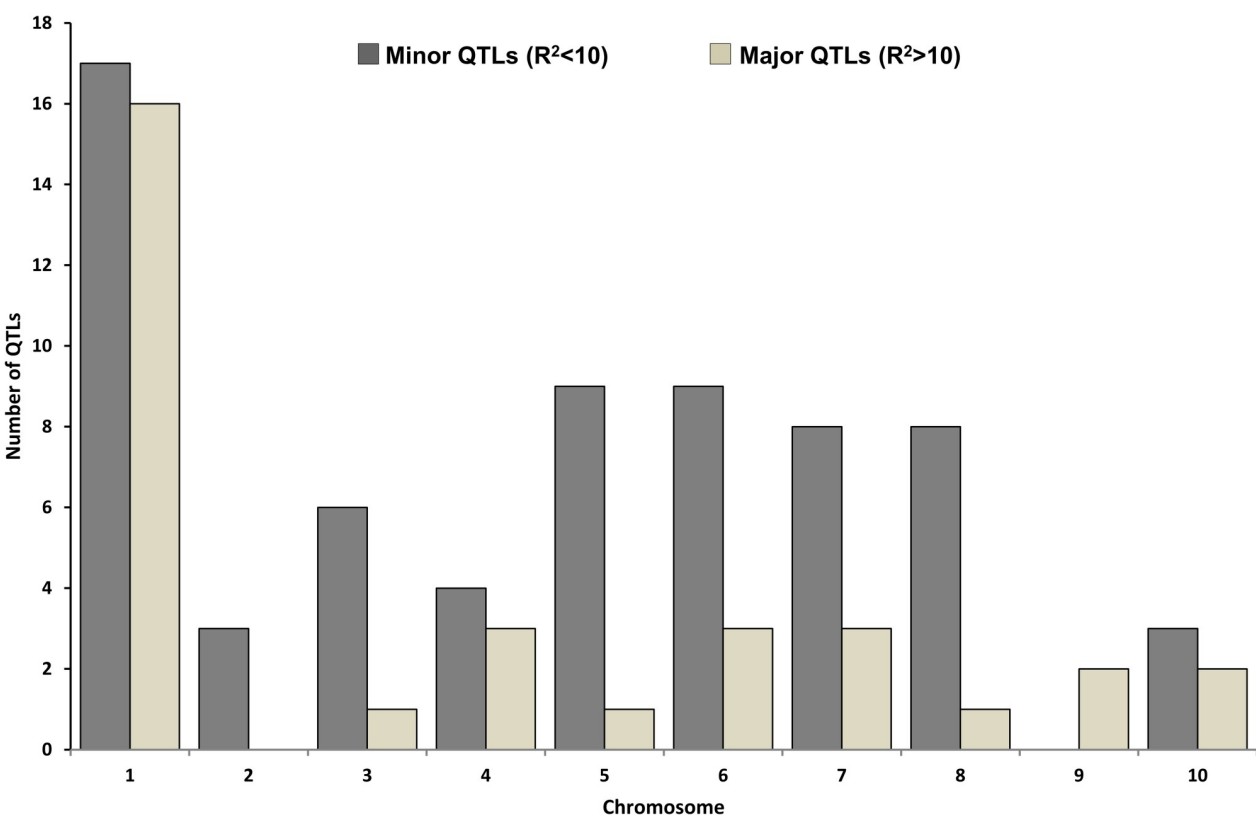

**Fig 2. Details of QTLs of the experiments used in meta-QTL analysis indicating total number of major QTLs ($R^2>10$) and minor QTLs ($R^2<10$).** Sixty seven out of the total 99 QTLs were minor, whereas rest of the thirty two QTLs were major. Indicating the control of large number of small effect QTLs and a few major effect QTLs and hence revealing the complex genetic architecture of popping traits.

number of small effect QTLs along with few major effect QTLs and hence, revealing the complex genetic architecture of popping traits.

## QTL projection and consensus map development

MetaQTL analysis was performed using the models with smallest AIC values. A total of ten metaQTLs were recognized which were distributed on two chromosomes, *viz.*, seven on chromosome 1 and three on chromosome 6 (Fig 3). Details of the ten metaQTLs identified are presented in Table 3. The average number of markers on chromosome 1 was 25.75 with average marker interval distance of 13.7 cM. In case of chromosome 6, the corresponding values for marker number and interval were 16.40 and 15.97 cM, respectively. Confidence interval (CI) of identified meta-QTLs was lower than their initial QTLs and ranged from 1.89 to 14.27 cM. The phenotypic variance among ten metaQTLs varied from 7.6% to 24.5%. The average recombination distance between marker intervals ranged from 90.99 to 2053.58 kb/cM on physical map, whereas, the coefficient of reduction in length from mean initial QTL to metaQTL varied from 0.53 to 4.29. The physical length ranged from 0.43 to 12.75 Mb. The reason behind the long physical interval was the unavailability of exact flanking markers. The physical lengths of the metaQTLs were searched using the 'locus pair lookup' browser (https://www.maizegdb.org/locus_pair_lookup). Nearly a thousand of genes were observed in the ten metaQTL regions. Using the 'q teller' tool at maize GDB a total of 229 genes were shortlisted on the basis of their expression in endosperm and pericarp tissue in ten metaQTLs (the

**Table 3. Details of the ten metaQTLs identified.**

| S. No. | Chr | Bin location | mQTLs | Position (cM) | Flanking markers | QTLs present in mQTL | Mean R² of QTLs | Range of R² QTLs | AIC | QTL model | Mean initial CI (cM) | mQTL CI (95%) | Physical length of mQTL (Mb) | Kb/cM | Coefficient of reduction | Candidate genes |
|---|---|---|---|---|---|---|---|---|---|---|---|---|---|---|---|---|
| 1 | 1 | 1.01 | mQTL1_1 | 26.80 | npi97a -T1-2 (4464) (1) | E2_qPR1/1, E3_qCT1-1, E2_qPV1/1, E3_qPV1-1 E3_qPR1-1 | 0.09 | 0.06–0.13 | 55.74 | 4 | 4.23 | 1.89 | 7.78–15.79 | 2053.58 | 2.24 | 48 |
| 2 | | 1.02 | mQTL1_2 | 33.90 | csu1190—csu691 | E3_qFS1-1, E2_qFS1/1 | 0.16 | 0.16 | | | 7.57 | 5.36 | 15.66–17.79 | 259.1 | 1.41 | 16 |
| 3 | | 1.02–1.03 | mQTL1_3 | 57.61 | hsp26—ms17 | E3_qCF1-1, 2_qPEV1-1 | 0.11 | 0.06–0.15 | | | 30.95 | 14.27 | 33.46–46.21 | 850.3 | 2.16 | 34 |
| 4 | | 1.07 | mQTL1_4 | 157.48 | T1-9b (1)—csu660a | E2_q100GW1-1, E4_qBPR1-1 | 0.09 | 0.08–0.09 | | | 17.99 | 10.82 | 214.92–217.33 | 187.1 | 1.66 | 9 |
| 5 | | 1.09 | mQTL1_5 | 192.01 | umc252b -umc197a (rip) | E3_qPR1-2, E2_qPR1/2, E2_qPEV1-2, E3_qPV1-2, E2_qPV1/2 | 0.08 | 0.07–0.10 | | | 14.19 | 6.35 | 257.38–262.94 | 732.28 | 2.23 | 21 |
| 6 | | 1.1 | mQTL1_6 | 205.87 | 1w1—bcd450b | E4_qBCP1-1, E3_qCT1-2, E4_qBCT1-1 | 0.11 | 0.08–0.13 | | | 7.27 | 2.27 | 272.93–275.48 | 771.99 | 3.2 | 15 |
| 7 | | 1.11 | mQTL1_7 | 247.27 | csu266—py2 | E2_qPR1/3, E2_qPV1/3 E3_qPR1-3, E4_qBPR1-2, E2_qPEV1-3, E3_qPV1-3 | 0.10 | 0.08–0.11 | | | 26.26 | 7.48 | 290.58–295.18 | 340.8 | 3.51 | 28 |
| 8 | 6 | 6.05 | mQTL6_1 | 106.84 | pdk1—umc152c | E5_qPR-6-2, E5_qPF-6-1 | 0.25 | 0.07–0.21 | 55.74 | 4 | 5.69 | 10.72 | 145.98–154.28 | 754.44 | 0.53 | 39 |
| 9 | | 6.06 | mQTL6_2 | 118.94 | umc138a -Php20904 | E2_qPR6/1, E2_qPEV6-1, E2_qPV6/1, 3_qCP6-1, E3_qPR6-1, E4_qBCP6-1, E5_qPF-6-1, E3_qPV6-1 | 0.10 | 0.06–0.21 | | | 8.87 | 2.07 | 156.46–158.59 | 344.29 | 4.29 | 10 |
| 10 | | 6.07 | mQTL6_3 | 152.20 | csu293—agp2 | E5_qPR-6-1, E5_qPV-6-1, E5_qPF-6-2 | 0.08 | 0.07–0.08 | | | 12.39 | 3.32 | 166.16–166.59 | 90.99 | 3.73 | 9 |

# For trait abbreviation refer to Table 1; Chr: Chromosome; CI: Confidence interval; R²: Regression coefficient; AIC: Akaike information criterion; kb: Kilobase; cM: Centimorgan.

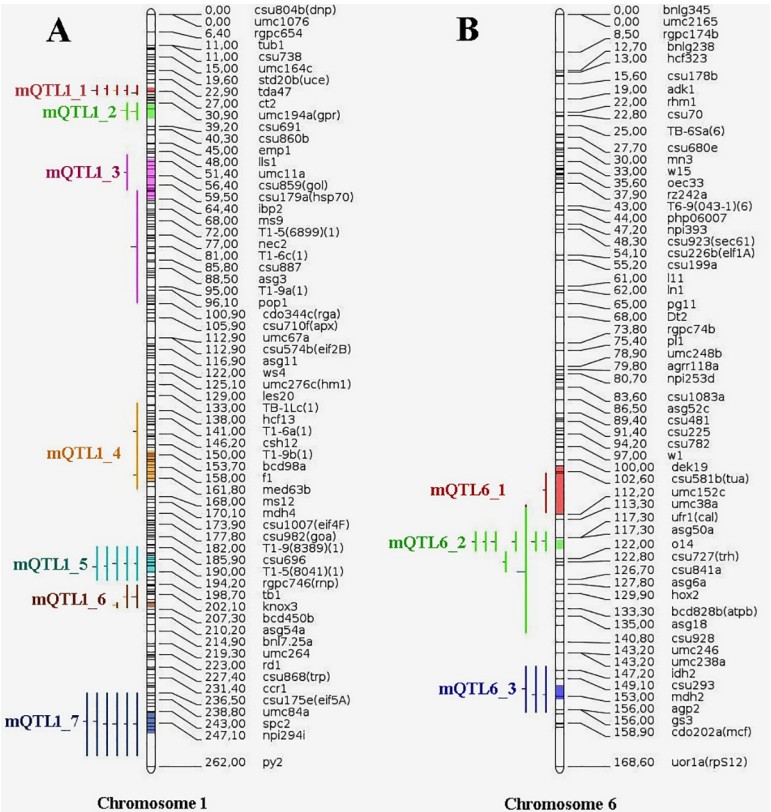

**Fig 3. Distribution of Ten metaQTLs for popping traits on chromosome 1 and 6.** (A) Seven metaQTLs, *viz.*, mQTL1_1, mQTL1_2, mQTL1_3, mQTL1_4, mQTL1_5, mQTL1_6 and mQTL1_7 identified on chromosome 1 with an average number of markers of 25.75 and average marker interval distance of 13.70 cM. (B) Three metaQTLs, *viz.*, mQTL6_1, mQTL6_2, mQTL6_3 were identified on chromosome 6 having the corresponding values for marker number and interval were 16.40 and 15.97cM, respectively.

threshold value taken was '10' for sorting genes expressing exclusively in pericarp and endosperm alone and simultaneously in both endosperm and pericarp) (S1 Table) (https://qteller. maizegdb.org/). A total of 94 candidate genes showed expression only in pericarp, such as– beta amylase-2 (Zm00001d027619), storage protein (Zm00001d035696), alpha/beta-Hydrolases superfamily protein (Zm00001d035570), geb7; glucan endo-1,3-beta-glucosidase7 (Zm00001d033423), Glycosyltransferase family 61 protein (Zm00001d038105) and others. Similarly, 20 candidate genes were observed to be expressed only in endospermic tissue, such as -nucleic acid binding protein (Zm00001d038506), protein spatula (Zm00001d034503, Tetratricopeptide repeat (TPR)-like superfamily protein (Zm00001d027613) etc. The identified 115 genes to be expressed in the pericarp and endospermic tissues such as—zein protein (Zm00001d035760), legumin1 (Zm00001d035700), expressed protein (Zm00001d033840) and others. Most of the genes identified were linked with storage proteins, transcription factors (TF) and defense related genes. Further, 19 out of 299 genes, *viz.*, Zm00001d027756, Zm00001d027861, Zm00001d027558, Zm00001d027612, Zm00001d027841, Zm00001d027619, Zm00001d027656, Zm00001d033753, Zm00001d033788, Zm00001d035595, Zm00001d032187, Zm00001d033328, Zm00001d033775, Zm00001d034460, Zm00001d034452, Zm00001d038142, Zm00001d038272, Zm00001d038193 and Zm00001d038508 were associated with metabolic pathways on the basis of Kyoto Encyclopedia of Genes and Genomes (KEGG). Out of the 229 genes, majorly the 19 genes linked with metabolic pathways need to

be further validated on the basis of loss or gain of function. Four out of ten metaQTLs identified in the study, *viz.*, mQTL1_1, mQTL1_5, mQTL1_7 and mQTL6_2 were considered important for popping as 5–8 QTLs across studies were clustered in these regions. Total 48 genes were identified in *mQTL1_1* on the basis of higher expression in endosperm and pericarp at 18 days after pollination (DAP). Some of the most important genes identified are Zm00001d027656, Zm00001d027619 and Zm00001d027675, expressing in pericarp only, whereas, genes Zm00001d027613 and Zm00001d027719 express solely in endosperm region. The genes expressing in both endosperm and pericarp included Zm00001d027558, Zm00001d027612, Zm00001d027841 etc.

Similarly, the genes expressing in the pericarp in *mQTL1_5* included Zm00001d033328, Zm00001d033375, Zm00001d033378 etc., whereas genes expressing in endosperm included Zm00001d033321, Zm00001d033337 and Zm00001d033447. The genes, *viz.*, Zm00001d033 291, Zm00001d033420, Zm00001d033388 etc. were identified to be expressed in both the endosperm and pericarp tissues. Genes identified in *mQTL1_7* with higher expression in pericarp were Zm00001d034452, Zm00001d034433, Zm00001d034497 and others, whereas, Zm00001d034503 and Zm00001d034440 genes expressed specifically in endosperm. The genes Zm00001d034460, Zm00001d034509, Zm00001d034432 and others expressed in both endosperm and pericarp tissues. Identified genes for *mQTL6_2* with pericarp specific expression were Zm00001d038532, Zm00001d038453, Zm00001d038529 etc. The genes Zm000 01d038506 and Zm00001d038467 were observed to express in endosperm region. The genes expressing in both pericarp and endosperm included Zm00001d038508, Zm00001d038525, Zm00001d038525 and others. The detailed information regarding remaining metaQTLs is provided in the Table 3.

## Discussion

The polygenic traits are affected by both environmental and genetic factors. As the popping traits are complex in nature, QTL mapping is the best approach to understand the genetic architecture of the concerned traits [18]. Genetic background, population size and genotype × environment interaction are crucial factors affecting the functionality of mapped QTLs. In case of large CI, several genes can be present and hence reliability of QTLs for use in MAS cannot be guaranteed based on few studies. Although, few studies led to the identification of major QTLs for popping traits, progress in deployment of these QTLs in maize breeding programme is limited because of lack of consistency and validation in variable environments and in new genetic backgrounds. In this direction identification of major effect QTLs related to popping traits across different environments and background is prerequisite to make advancement through marker assisted breeding (MAB). Meta-analysis of QTLs provides an opportunity to compile QTL information from different studies to get more accurate location of QTLs. It is an effective tool for optimization and validation of known QTLs by shrinking the confidence interval, and identifying the true QTLs via accurate consensus QTLs called as meta-QTLs [19]. Though meta-analysis for popping related QTLs involving three populations evaluated in four environments were reported [6], a comprehensive study on published information is lacking, which was addressed in the current study.

In this study, 99 reported QTLs from eight studies were projected on consensus map of maize genome. A total of ten metaQTLs were identified. The metaQTLs, *viz.*, mQTL1_1, mQTL1_5, mQTL1_7 and mQTL6_2 had five to eight QTLs clustered in the regions, hence were considered important for popping traits. The mQTLs identified in this study were compared with the earlier reports. mQTL1_1 at bin location 1.01 consisted of five QTLs related to three traits, *viz.*, PR, CT and PV. MetaQTL analysis for popping traits identified metaQTL at

the same region (bin 1.01) containing seven QTLs for traits like PV, PR and PF [13]. QTLs related to various agronomic traits like stalk diameter, tassel length, leaf area and plant height are also clustered in this region [20]. Tropical popcorn lines are characterized by weak stalk, large tassel and small ears [21]. Since, major QTLs for popping related traits are co-localized with stalk diameter, tassel length, leaf area and plant height in this region these traits can simultaneously be improved while introgressing this metaQTL. In the current study, another metaQTL has been identified at bin locus 1.09 involving five QTLs related to three different traits, *viz.*, PR, PV and PEV which is consistent withearlier reported metaQTL at bin location 1.08–1.10 consisting of 10 QTLs for PF, PV and PR [13]. Another major effect mQTL 1_7 consisting of six QTLs related to PR, PV and PEV was found at 1.11 bin locus. In corroboration with earlier spotted metaQTL with 11 QTLs for traits like PF, PV and PRat bin region 6.06–6.07 [13]. Current study also identified two metaQTLs (mQTL6_2 and mQTL6_3) with eight and three QTLs at bin location 6.06 and 6.07, respectively for five popping traits, *viz.*, PR, PEV, PV, PF and CP. Hence, the results of this metaQTL analysis study are in agreement with earlier findings [20]. Interestingly, QTLs related to root traits, *viz.*, primary root length, seminal root number, crown root length, lateral root length and lateral root number have also been reported in the same region [22]. This probably indicates towards the complexity of plant architecture of popcorn as similar genomic regions can govern expression of entirely different traits like in case of pleiotropic effects [23]. The meta-QTLs with a steady and large effect on target trait and with small physical and genetic interval can potentially be used in MAS. The percent phenotypic variance explained by the QTL is an effective parameter to decide the worthiness of QTL for use in marker assisted breeding programmes [24]. The meta-QTLs identified in current study explained quite a high phenotypic variance in range of 7.6% to 24.5%.

The four major metaQTLs namely metaQTL1_1, metaQTL1_5, metaQTL1_7 and metaQTL6_2 are considered important for popping traits as 5–8 QTLs were clustered in these regions with moderately high $R^2$ value. The clustered QTLs were also from two (metaQTL1_1 and metaQTL1_5) and more than two (metaQTL1_7: 3 and metaQTL6_2: 4) different experiments. Further, out of the 19 genes specifically involved in carbohydrate metabolism, eleven are residing in these regions. Hence, these metaQTLs with higher PEV can serve as potential targets for further exploration in popcorn improvement. Gene annotation programme at maizeGDB revealed presence of large number of genes in the 10 metaQTL regions. Since, popping traits are more influenced by the properties of endosperm and pericarp, targeted search was performed for genes specifically expressing in endosperm and pericarp. Using the tool 'qTeller', the number of genes expressing in pericarp and endosperm tissues was narrowed down to 229. In above mentioned four metaQTLs, some of the potential genes with higher expression in endosperm and pericarp at 18 DAP in B73 (the maize line first fully sequenced)are: Zm00001d027667 (citt1;citrate transporter1), Zm00001d027756 (12-dihydroxy-3-keto-5-methylthiopentene dioxygenase 3), Zm00001d027595 (60S ribosomal protein L23), Zm00001d033371(Dual specificity protein phosphatase 1B), Zm00001d033291 (Probable WRKY transcription factor 74), Zm00001d034422 (40S ribosomal protein S18), Zm00001d034505 (Vesicle-associated membrane protein 722), Zm00001d034429 (PITH domain-containing protein), Zm00001d038519 (hypothetical protein [Zea mays]) etc. Further, 19 genes were selected on the basis of carbohydrate metabolism in the KEGG pathways which may contribute to the popping traits in popcorn. The validation of these genes needs to be carried out on the basis of loss or gain of function leading to popping expansion. In earlier study also, SNPs linked to popping expansion were reported and annotated genes related to the associated SNPs were identified along with the functions associated with starch content, playing important role in popping expansion. The SNP S2_1307940234 was identified for popping expansion and was located inside the gene model AC212835.3_FG001 which was related to a

Pentatricopeptide Repeat (PPR) protein [25]. PPR proteins were reported to affect amylose biosynthesis and also for tocopherol contents in maize kernels [26]. Another SNP, S3_1223876449 with gene model GRMZM2G038170 was also reported but without any identified protein function [25]. These metaQTL regions can serve as important regions for fine mapping and candidate gene finding. The QTLs reported in these regions can be used in MAS programme for introgression of popping associated traits in the popcorn lines to enhance the popping ability.

## Conclusion

Identification of accurate and consistent QTLs across different genetic backgrounds and environments is necessary to utilize the identified QTLs in marker-assisted breeding programme. In this regard ten metaQTLs have been identified harbouring major consensus QTLs governing popping traits in maize using the Akaike information criterion (AIC) based method. Four out of ten metaQTLs were considered important as they harboured clusters of 5–8 QTLs governing the popping traits. The available genes in the metaQTL regions were analysed using the gene annotation information available in the maizeGDB. Genes were shortlisted on the basis of their expression in endosperm and pericarp tissues. The Kyoto Encyclopedia of Genes and Genomes (KEGG) pathways analysis which serves as a resource database for the insight of utilities and functions of biological system was used to further select the genes on the basis of carbohydrate metabolism. The identified genes could further be validated on the basis of loss or gain of function and used in the marker assisted selection for introgression of popping traits towards enhancing the popping ability.

## Supporting information

**S1 Checklist. PRISMA 2009 checklist.**
(DOC)

**S1 Table. Identified genes in metaQTL regions based on expression in endosperm and pericarp.**
(XLSX)

**S2 Table. Features of the individual QTL related to popping traits in maize reported in different studies describing the name of QTL, position in the map, confidence interval and R$^2$ value for each QTL.**
(XLSX)

## Acknowledgments

Authors acknowledge the inputs of Dr. Shelly Parveen, Indian Agricultural Research Institute, New Delhi (India) for discussion on biochemical interpretation. The first author thanks Punjab Agricultural University (PAU), Ludhiana for the logistic support during course of investigation.

## Author Contributions

**Formal analysis:** Sukhdeep Kaur, Mukesh Choudhary, Abhijit Kumar Das, Ranjeet Ranjan Kumar.

**Methodology:** Sukhdeep Kaur.

**Resources:** Sujay Rakshit.

**Software:** Sukhdeep Kaur, Mukesh Choudhary.

**Supervision:** Sujay Rakshit.

**Writing – original draft:** Sukhdeep Kaur.

**Writing – review & editing:** Sujay Rakshit, Mukesh Choudhary, Abhijit Kumar Das.

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
