## [Decision Letter · Decision Letter 0]

25 Jan 2021

PONE-D-20-37524

Meta-analysis of QTLs associated with popping traits in maize (Zea mays L.)

PLOS ONE

Dear Dr. Rakshit,

Thank you for submitting your manuscript to PLOS ONE. After careful consideration, we feel that it has merit but does not fully meet PLOS ONE’s publication criteria as it currently stands. Therefore, we invite you to submit a revised version of the manuscript that addresses the points raised during the review process.

We look forward to receiving your revised manuscript.

Kind regards,

Ajay Kumar

Academic Editor

PLOS ONE

2. In your Methods section, please ensure you have included all the details of the search strategy for literature review, i.e. search keywords/date/language, inclusion/exclusion criteria, etc. for reproducibility purpose.

"The work was carried out under the financial assistance from the Indian Council of

346 Agricultural Research, New Delhi (India). Inputs of Dr. Shelly Parveen, Indian Agricultural Research Institute,

347 New Delhi (India) for discussion on biochemical interpretation. The first author thanks Punjab Agricultural

348 University for the logistic support during course of investigation."

Reviewers' comments:

Reviewer's Responses to Questions

**Comments to the Author**

1. Is the manuscript technically sound, and do the data support the conclusions?

Reviewer #1: Yes

Reviewer #2: Yes

2. Has the statistical analysis been performed appropriately and rigorously? 

Reviewer #1: Yes

Reviewer #2: Yes

3. Have the authors made all data underlying the findings in their manuscript fully available?

Reviewer #1: Yes

Reviewer #2: Yes

4. Is the manuscript presented in an intelligible fashion and written in standard English?

Reviewer #1: Yes

Reviewer #2: Yes

5. Review Comments to the Author

Reviewer #1: Minor comments:

1- Abbreviations should be explained in their very first appearance in the text, please explain these abbreviations: QTL (line 9), CGAR (line 28), psi (line 39), PEV (line 54), DAP ( line 320), and describe what is B73 (line 329).

2- Line 180, there should be space in “tenmetaQTL”

3- Line 206- 271: As describing the “The features of the four major metaQTLs identified in the study” no need of describing the features of the metaQTLs that already mentioned in the Table 3. Most of the features already mentioned in Table 3.

4- You could add a supplementary table describing the features of the individual QTL from different studies that used in this article.

Reviewer #2: Summary:

With the goal to compile QTLs for popping-related traits, the authors collected 99 QTLs from eight studies. They came up with 10 meta-QTLs on chromosomes 1 and 6. They related these metaQTLs to 229 candidate genes that expressed in pericarp, endosperm, or both. Finally, this study revealed consensus metaQTLs for popping traits, which the authors recommended for markers-assisted breeding. This study has the merit of identifying consensus genomic regions that could be important for popping quality traits. However, I will pinpoint some of the missing things in the paper and my comments/suggestions as follows:

1. Only papers published between 2006 and 2012 are included for the metaQTL analysis. It is not clear why you did not included papers after 2012. At lease, I find one paper in 2021 "Thakur, S., Kumar, R., Vikal, Y. et al. Molecular mapping of popping volume QTL in popcorn (Zea maize L.). J. Plant Biochem. Biotechnol. (2021). " ext-link-type="uri" xlink:type="simple">https://doi.org/10.1007/s13562-020-00636-y". I think it needs some justification for not including papers published after 2012.

2. There is inconsistency in citing references in text. You used both "number" and "author-date" styles. Your citations in the text need to be consistent as recommended by the journal.

3. The Figures are poor in quality and I think they need to be improved in the final submission.

4. I think it is good to avoid using "etc" in listing things (example on line 19: "MetaQTL1_1 at bin location 1.01 coincided with the reported QTLs related to various agronomic traits like stalk diameter, tassel length etc."). If the list is not big enough, I think it is better to mention all of them. This works for the other "etc" mentioned in the text.

5. Abbreviations must be defined first before they can be used the next time it comes in the text. (examples: "PEV" on line 54 and "PVE" on line 81).

6. The discussion part: The first part (paragraphs 1 and 2) is discussed in comparison with past research or generally describing concept or facts referring to literature. I think this is good. The second part (paragraph 3) seems a repetition of the results part. I suggest you look into the discussion part and try to improve it.

7. Line 204-206: You declared 4 metaQTLs important because they were represented by 5-8 QTLs. Is this enough to classify some as important and the others as not important. I have not seen any other criteria for your decision to classify as important and not important. I think this needs some explanation. I think some other criteria should be added in addition to the number of QTLs. This simple classification may sideline some of the metaQTLs which can be important in breeding.

8. Overall, I suggest a through review of the manuscript according to the guideline of the journal.

6. PLOS authors have the option to publish the peer review history of their article (what does this mean?). If published, this will include your full peer review and any attached files.

Reviewer #1: No

Reviewer #2: No

---

## [Author Response · Author response to Decision Letter 0]

26 Mar 2021

The response to reviewer file has been submitted separately which address each and every points raised by the reviewers.

---

## [Decision Letter · Decision Letter 1]

28 Apr 2021

PONE-D-20-37524R1

Meta-analysis of QTLs associated with popping traits in maize (Zea mays L.)

PLOS ONE

Dear Dr. Rakshit,

Thank you for submitting your manuscript to PLOS ONE. After careful consideration, we feel that it has merit but does not fully meet PLOS ONE’s publication criteria as it currently stands. Therefore, we invite you to submit a revised version of the manuscript that addresses the points raised during the review process.

If applicable, we recommend that you deposit your laboratory protocols in protocols.io to enhance the reproducibility of your results. Protocols.io assigns your protocol its own identifier (DOI) so that it can be cited independently in the future. For instructions see: http://journals.plos.org/plosone/s/submission-guidelines#loc-laboratory-protocols. Additionally, PLOS ONE offers an option for publishing peer-reviewed Lab Protocol articles, which describe protocols hosted on protocols.io. Read more information on sharing protocols at https://plos.org/protocols?utm_medium=editorial-emailutm_source=authorlettersutm_campaign=protocols.

We look forward to receiving your revised manuscript.

Kind regards,

Ajay Kumar

Academic Editor

PLOS ONE

Journal Requirements:

Reviewers' comments:

Reviewer's Responses to Questions

**Comments to the Author**

1. If the authors have adequately addressed your comments raised in a previous round of review and you feel that this manuscript is now acceptable for publication, you may indicate that here to bypass the “Comments to the Author” section, enter your conflict of interest statement in the “Confidential to Editor” section, and submit your "Accept" recommendation.

Reviewer #1: All comments have been addressed

Reviewer #2: All comments have been addressed

2. Is the manuscript technically sound, and do the data support the conclusions?

Reviewer #1: (No Response)

Reviewer #2: Yes

3. Has the statistical analysis been performed appropriately and rigorously? 

Reviewer #1: Yes

Reviewer #2: Yes

4. Have the authors made all data underlying the findings in their manuscript fully available?

Reviewer #1: Yes

Reviewer #2: Yes

5. Is the manuscript presented in an intelligible fashion and written in standard English?

Reviewer #1: (No Response)

Reviewer #2: No

6. Review Comments to the Author

Reviewer #1: Line 117:” Lod value” Lod should be capitalized.

Line 128, table 2: Row 2 and 3, in Reference column, there is no need to add semicolon after references. row 5 in reference column ” ) [7, 24]”, there is a redundant parenthesis. Also, at the footnote of table 2 describe what is RIL, BC and F mapping population.

Line 185: write the reference of the “q teller”.

Line 124, and 210 : Fig 2 and 3: All the caption of figures should be Bold.

Line 78, 128 and 215: Table, 1,2, 3 captions font are not consistent with the figures caption font , they are smaller.

Line 312: There is a redundant parenthesis at the end of sentences .”Features of the individual QTL related to popping traits in maize reported in different studies)”

General comment: Fig 2 and Fig 3 do not have good quality and resolution.

Reviewer #2: All comments except one addressed satisfactorily. I still prefer some recent studies to be included. But the authors preferred to keep 8 studies for their Meta-QTL analysis. That could be fine and recent QTL studies could be addressed in a different study. I prefer listing all the items without using “etc”. If the lists are too long, it is possible to use the word “including” followed by listing of the first few items. That is just personal preference. Overall, I can say the review comments addressed.

Some other changes that I want to suggest are as follows:

It seems rephrasing sentences necessary in several places in the manuscript:

Example1--- Line 16-17: The clustered QTLs were from ≥2 different experiments and out of the 19 genes specifically involved in carbohydrate metabolism, eleven were in these regions thus considered as most important.

Could be re-written as: The clustered QTLs were from two or more experiments. Of the 19 genes specifically involved in carbohydrate metabolism, 11 of them were in these regions, implying the importance of these clustered QTLs.

Example2--- Line 18-19: A total of 229 genes were shortlisted in these regions on the basis of their expression in endosperm and pericarp tissues using qteller option in maizeGDB, out of which 19 were specifically involved in carbohydrate metabolism.

Could be re-written as: Based on the expression pattern in endosperm and pericarp tissues, a total of 229 genes were selected. Nineteen of these genes involves in carbohydrate metabolism.

In general, it may be good to go through the manuscript and make changes as required.

7. PLOS authors have the option to publish the peer review history of their article (what does this mean?). If published, this will include your full peer review and any attached files.

Reviewer #1: No

Reviewer #2: No

---

## [Author Response · Author response to Decision Letter 1]

22 May 2021

The response to reviewers file has been attached which explain the queries raised.

---

## [Decision Letter · Decision Letter 2]

29 Jun 2021

PONE-D-20-37524R2

Meta-analysis of QTLs associated with popping traits in maize (Zea mays L.)

PLOS ONE

Dear Dr. Rakshit,

Thank you for submitting your manuscript to PLOS ONE. There are few minor revisions still required. Therefore, we invite you to submit a revised version of the manuscript that addresses the points raised during the review process.

If applicable, we recommend that you deposit your laboratory protocols in protocols.io to enhance the reproducibility of your results. Protocols.io assigns your protocol its own identifier (DOI) so that it can be cited independently in the future. For instructions see: http://journals.plos.org/plosone/s/submission-guidelines#loc-laboratory-protocols. Additionally, PLOS ONE offers an option for publishing peer-reviewed Lab Protocol articles, which describe protocols hosted on protocols.io. Read more information on sharing protocols at https://plos.org/protocols?utm_medium=editorial-emailutm_source=authorlettersutm_campaign=protocols.

We look forward to receiving your revised manuscript.

Kind regards,

Ajay Kumar

Academic Editor

PLOS ONE

Journal Requirements:

Reviewers' comments:

Reviewer's Responses to Questions

**Comments to the Author**

1. If the authors have adequately addressed your comments raised in a previous round of review and you feel that this manuscript is now acceptable for publication, you may indicate that here to bypass the “Comments to the Author” section, enter your conflict of interest statement in the “Confidential to Editor” section, and submit your "Accept" recommendation.

Reviewer #1: All comments have been addressed

Reviewer #2: All comments have been addressed

2. Is the manuscript technically sound, and do the data support the conclusions?

Reviewer #1: Yes

Reviewer #2: Yes

3. Has the statistical analysis been performed appropriately and rigorously? 

Reviewer #1: Yes

Reviewer #2: Yes

4. Have the authors made all data underlying the findings in their manuscript fully available?

Reviewer #1: Yes

Reviewer #2: Yes

5. Is the manuscript presented in an intelligible fashion and written in standard English?

Reviewer #1: Yes

Reviewer #2: Yes

6. Review Comments to the Author

Reviewer #1: 1- The Fig 2 and 3 still do not have good quality and resolution. In Fig 3, name of the markers should be clear.

2- Make sure that Table 3 is in good shape:

a) It has one redundant row at the bottom, that should be deleted,

b) Choose brief header, and make the font of the headers and the content of the Table smaller (for example 7)

until they can be fit in on or two lines,

c) All digits in Table 3 with decimal should be rounded to 1 or 2 decimal places, be consistent all over the digits.

d) Explain what chr, CI, R2, AIC, kb, and cM are at the footnote of Table.

Reviewer #2: The authors addressed all the comments satisfactorily. I believe the manuscript is acceptable for publication.

7. PLOS authors have the option to publish the peer review history of their article (what does this mean?). If published, this will include your full peer review and any attached files.

Reviewer #1: No

Reviewer #2: No

---

## [Author Response · Author response to Decision Letter 2]

8 Jul 2021

The reviewer comments have been addressed in the reviewer response file.

---

## [Editor Report · Decision Letter 3]

6 Aug 2021

Meta-analysis of QTLs associated with popping traits in maize (Zea mays L.)

PONE-D-20-37524R3

Dear Dr. Rakshit,

We’re pleased to inform you that your manuscript has been judged scientifically suitable for publication and will be formally accepted for publication once it meets all outstanding technical requirements.

Kind regards,

Ajay Kumar

Academic Editor

PLOS ONE
---

## [Editor Report · Acceptance letter]

11 Aug 2021

PONE-D-20-37524R3 

Meta-analysis of QTLs associated with popping traits in maize (*Zea mays* L.) 

Dear Dr. Rakshit:

I'm pleased to inform you that your manuscript has been deemed suitable for publication in PLOS ONE. Congratulations! Your manuscript is now with our production department. 

Kind regards, 

on behalf of

Dr. Ajay Kumar 

Academic Editor

PLOS ONE